# ZERO-LEVEL-SET ENCODER FOR NEURAL DISTANCE FIELDS

## ABSTRACT

Neural shape representation generally refers to representing 3D geometry using neural networks, e.g., to compute a signed distance or occupancy value at a specific spatial position. Previous methods tend to rely on the auto-decoder paradigm, which often requires densely-sampled and accurate signed distances to be known during training and testing, as well as an additional optimization loop during inference. This introduces a lot of computational overhead, in addition to having to compute signed distances analytically, even during testing. In this paper, we present a novel encoder-decoder neural network for embedding 3D shapes in a single forward pass. Our architecture is based on a multi-scale hybrid system incorporating graph-based and voxel-based components, as well as a continuously differentiable decoder. Furthermore, the network is trained to solve the Eikonal equation and only requires knowledge of the zero-level set for training and inference. Additional volumetric samples can be generated on-the-fly, and incorporated in an unsupervised manner. This means that in contrast to most previous work, our network is able to output valid signed distance fields without explicit prior knowledge of non-zero distance values or shape occupancy. In other words, our network computes approximate solutions to the boundary-valued Eikonal equation. It also requires only a single forward pass during inference, instead of the common latent code optimization. We further propose a modification of the loss function in case that surface normals are not well defined, e.g., in the context of non-watertight surface-meshes and non-manifold geometry. Overall, this can help reduce the computational overhead of training and evaluating neural distance fields, as well as enabling the application to difficult shapes. We finally demonstrate the efficacy, generalizability and scalability of our method on datasets consisting of deforming 3D shapes, single class encoding and multiclass encoding, showcasing a wide range of possible applications.

## 1 INTRODUCTION

Algorithms processing 3D geometric data have become omnipresent and an integral part of many systems. These include for example the systems evaluating Lidar sensor data, game engine processing, visualizing 3D assets and physical simulation used in engineering prototypes. In recent years, deep learning methods have been increasingly investigated to assist in solving problems pertaining to 3D geometry.

In particular, neural shape representation deals with using neural networks to predict shape occupancies or surface distances at arbitrary spatial coordinates. Recent works have shown the ability to capture intricate details of 3D geometry with ever increasing fidelity (Wang et al., 2022). However, a significant number of these works employ an auto-decoder based architecture, which requires solving an optimization problem when representing new geometry. Additionally, the auto-decoder still has to be evaluated for all query points individually, which can become very costly when evaluating these systems for high-resolution reconstruction (Xie et al., 2022). Finally, many of these methods also require annotated and densely sampled ground truth data. Meta learning approaches, e.g., by Sitzmann et al. (2020a); Ouasfi & Boukhayma (2022), mitigate this problem, but also have to run several iterations of gradient descent to specialize the network for each new model before inference. Encoder-decoders can instead encode the shape in a single forward pass, and typically employ com-

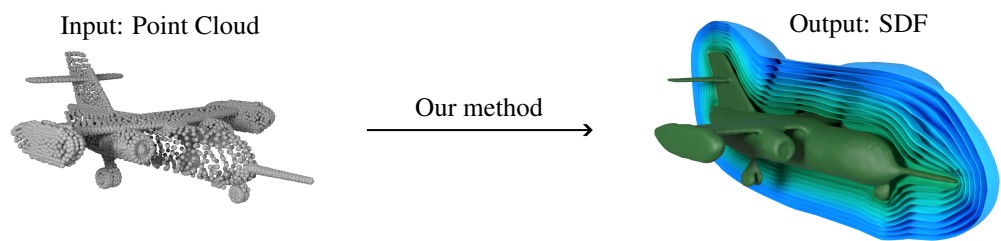

Figure 1: A schematic overview of our end-to-end trainable system. When given a surface point cloud, our neural network can directly compute the signed distance at a query point.

putationally cheaper decoder networks for evaluating query points. Nevertheless, these approaches also often require elaborate data preprocessing pipelines, and rely on labeled training data.

To address these issues, we propose an end-to-end learnable encoder-decoder system, that is not bound by previous data preprocessing constraints and can be trained using only the zero-level set, i.e., surface samples as labeled data. This kind of training was previously introduced for auto-decoders, and is enabled by using the Eikonal equation as a training target, yet remains, to the best of our knowledge, unexplored using encoders.

We summarize our contributions as follows: 1) We propose a novel encoder architecture for representing 3D shapes. The key component of this encoder is the hybrid and interleaved execution of graph-level convolutions and 3D grid convolutions. Our motivation for this hybrid approach is the ability to propagate information both in surface-point (geodesic) space and grid (Euclidian) space. 2) We show that the accuracy of our method is also intuitively controllable. Using a model with as little as 38K parameters (including the decoder) can already achieve excellent visual quality while being very fast to evaluate. This makes it useful for practical applications within resource constrained computing platforms. 3) The introduction of training encoder-decoder networks for 3D shapes on the Eikonal equation. This avoids having to compute *any* ground truth signed distance samples for training the network, while the output of the network is trained to exhibit the properties of a valid signed distance field (SDF). We also propose a simple modification to the loss function that can gracefully handle poorly oriented surface normals in the training data, caused by non-manifold or non-watertight geometry.

In the evaluation we show, that we are able to reconstruct better quality surfaces than other state-of-the-art methods. A schematic overview of our method is shown in Figure 1.

## 2    RELATED WORK

**Neural fields**    have become an integral part of research in geometric deep learning, with hundreds of papers published in recent years. A comprehensive overview is given by Xie et al. (2022). One of the seminal works on deep learning for unstructured data was the introduction of PointNet (Qi et al., 2017). From todays perspective, one of the major limitations of this work is the difficulty of learning high-frequency functions from low-dimensional data (Xu et al., 2019; Rahaman et al., 2019). The solution to the problem is addressed by more recent approaches such as NeRFs (Mildenhall et al., 2020) and Fourier Feature Networks (Tancik et al., 2020). In essence, the idea is to use positional embeddings, inspired by the embeddings proposed by Vaswani et al. (2017) for transformer networks. These embeddings compute a mapping from low dimensional positional information (typically 2D or 3D) into higher dimensional spaces using a specific number of Fourier basis functions (Tancik et al., 2020). A concurrent work shows that using periodic activation functions inside a MLP also significantly improves reconstruction quality and surface detail (Sitzmann et al., 2020b), a single layer of which can again be seen as a kind of positional encoding (Xie et al., 2022). Subsequent works improve the usage of positional encodings, e.g., by controlling the frequency through a feedback loop (Hertz et al., 2021) or modulating the periodic activations using a separate ReLU activated MLP (Mehta et al., 2021). Other benefits of using periodic activations are the ability to better learn high-frequency mappings and the continuous differentiability of these activations which is useful for evaluating training targets (Chen et al., 2023; Wang et al., 2022; Raissi et al., 2019).

There are many approaches for representing 3D shapes using neural networks. For clarity of exposition we will classify them into global and local methods.

**Global methods** do not make use of geometric structures in the network itself, and can generally be used irrespective of the discretized representation of the geometry. Typically auto-decoder methods, in which the latent representation is optimized during training and testing, are in this category (Sitzmann et al., 2020b; Wang et al., 2022; Park et al., 2019; Mescheder et al., 2019; Mehta et al., 2021; Gropp et al., 2020; Atzmon & Lipman, 2020b;a). The network can then be queried using both the latent feature and a 3D coordinate, to evaluate either a distance metric, or an occupancy value. Meta learning approaches also fall into this category. A few iterations of gradient descent are used to specialize the weights of a generalized network to a new shape (Sitzmann et al., 2020a; Ouasfi & Boukhayma, 2022). An approach that has both discretization-dependent and independent components was presented by Chen et al. (2023), where the discretization-dependent encoder is typically discarded during inference. The amount of encoded details by these methods is naturally bounded by the number of network weights. It has also been shown that using pooling-based set encoders for global conditioning frequently underfits the data (Buterez et al., 2022).

**Local methods** typically rely on using spatial structures within the network itself for the extraction of meaningful information (Peng et al., 2020; Chabra et al., 2020; Jiang et al., 2020; Chibane et al., 2020; Lombardi et al., 2019; Boulch & Marlet, 2022). This has proven to be a valuable approach, since it is quite difficult for neural networks to encode the high-frequency functions needed to represent detailed fields in 3D. Previous works make effective use of discretized structures, e.g., point-clouds, meshes or voxel-grids as either inputs or outputs (Qi et al., 2017; Hertz et al., 2021; Buterez et al., 2022; Takikawa et al., 2021). For encoder-type methods, extracting local features has been shown to improve network performance over global ones. As a kind of local conditioning, our approach falls into this category.

**Unsupervised SDF training** has been explored in some recent works (Atzmon & Lipman, 2020a;b; Sitzmann et al., 2020b; Gropp et al., 2020), however, with a focus on auto-decoders and generally using optimization of latent codes during inference. Notably, Gropp et al. (2020) introduced the formulation of the unsupervised Eikonal loss, which was further refined in the work of Sitzmann et al. (2020b). In our work we extend this loss to improve training on data with inconsistent normal orientations.

## 3 ZERO-LEVEL-SET ENCODER

In the following we will present our encoder-decoder architecture, including the newly introduced *convolution block* for the encoder, the decoder structure, the loss function along with our modification to support non-manifold geometries, and the overall training process. Our inputs are surface point-clouds of 3D models, given by a set of vertices $\mathcal{V} = \{V \in \mathbb{R}^3\}$. In order to use graph convolutions, we create edges $\mathcal{E} = \{E \in (\mathbb{N} \times \mathbb{N})\}$ between vertices, using e.g. k-nearest-neighbor (k-NN) or radius graph connectivity. Within the network, the surface points store abstract $f$-dimensional feature vectors ($V^f \in \mathbb{R}^f$), rather than 3D coordinates. This input representation also allows utilization of known mesh connectivity, e.g. for partially meshed point cloud inputs or triangle soups.

### 3.1 ENCODER-DECODER ARCHITECTURE

**Convolution block.** We introduce a novel hybrid graph-grid convolution block (shown in Figure 2) as the main building block for our encoder. In contrast to many previous encoder approaches that use either only point data (Qi et al., 2017; Mescheder et al., 2019) or transfer point data to voxel grids for further processing (Peng et al., 2020; Chibane et al., 2020), we instead interleave these approaches.

A graph convolution operator (as described by Wang et al. (2019)) transforms each vertex $V^f$ using the edge connectivity information. This transformation is invariant towards permutations of the vertex and edge lists. Next, the feature vectors are projected onto a grid via element-wise max pooling from vertices located in that grid cell. A $2 \times 2 \times 2$ convolution and subsequent deconvolution (where the feature count is doubled for the latent space to retain information) is then used to transfer

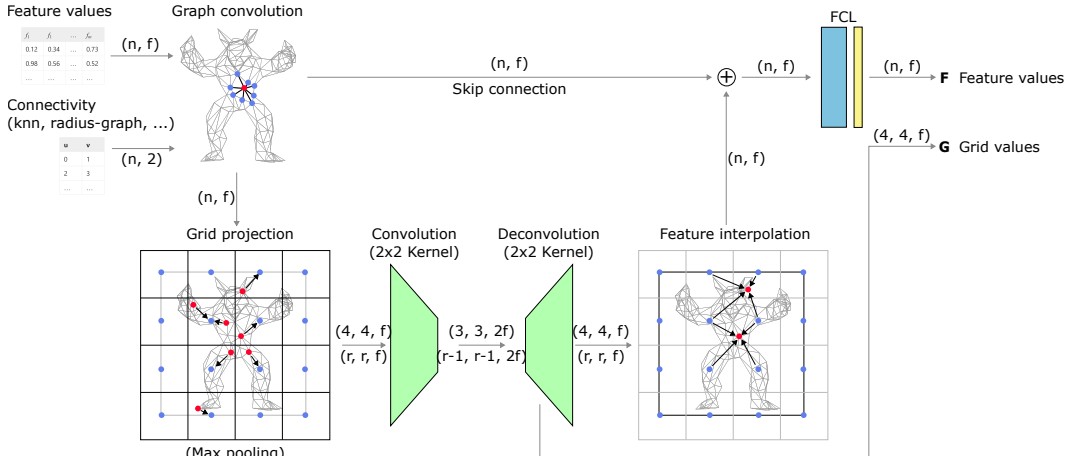

Figure 2: Convolution block that extracts features for a specific grid resolution. For clarity of illustration, a 2D rather than 3D grid is shown here. The input is a set of vertices (with position / feature data) and edges (encoded as vertex indices). The + denotes element-wise vector addition. The block has two outputs, feature values on the vertices and grid values for each grid cell. For all resolutions, a $2 \times 2$ convolution kernel is used. $n$: number of vertices. $f$: number of features (on the first level, the features are the spatial coordinate of each vertex). $r$: grid resolution.

information between neighboring cells. The features are then mapped back onto the vertices through tri-linear interpolation using the 8 closest cell centers. Here, they are combined with the original output of the graph convolution before finally being processed through a single per-vertex fully-connected layer. This output serves as the input of the next convolution block, while the deconvolved grid values are cached for later use.

The intuition behind our architecture is that the grid is suitable for distances in Euclidian space, while the graph convolution can extract information in the geodesic space of the object surface. Combining both multiple times throughout the network makes the feature extraction time and memory efficient.

**Encoder.** The overall encoder-decoder architecture is shown in Figure 3. At the beginning of the encoder, the input vertex positions $\mathcal{V}$ are transformed to per-vertex features $\mathcal{V}^f$ through positional encoding as described by Tancik et al. (2020) and a single linear layer with ReLU activations. The results are fed through a fixed number (typically 4) of our hybrid convolution blocks (as described above) with increasing resolution (contrary to the usual decreasing order of convolutional neural networks). The concatenated outputs of the grids form the grid vector (the feature vertex output of the last convolutional block is discarded). Values from each level of the grid vector are extracted for each of the $s$ SDF sample positions by tri-linear interpolation. The results are summed element-wise to form the final latent vector. We note that when latent vectors for additional sample positions are computed, there is no need to recompute the grid vector — it can be cached for future use.

**Decoder.** The decoder receives the sample features and positions as input and processes them individually to compute a single signed distance value for each sample. As architecture we make use of the proposed decoder of Mehta et al. (2021), which is an augmentation of the SIREN layer proposed by Sitzmann et al. (2020b). The feature vector is passed through a ReLU-activated MLP and the sample position through a Sine-activated MLP. The activations of the ReLU MLP are then used to modulate the activations of the Sine MLP whose output is the final SDF sample.

## 3.2 LOSS FUNCTION

An SDF can be defined as the unique solution $\Phi$ of the following Eikonal equation:

$$\begin{aligned} |\nabla\Phi(\boldsymbol{x})| &= 1 \text{ for } \boldsymbol{x} \in \Omega \setminus \Omega_S \subset \mathbb{R}^3 \\ \Phi(\boldsymbol{x}) &= 0 \text{ for } \boldsymbol{x} \in \Omega_S, \end{aligned} \tag{1}$$

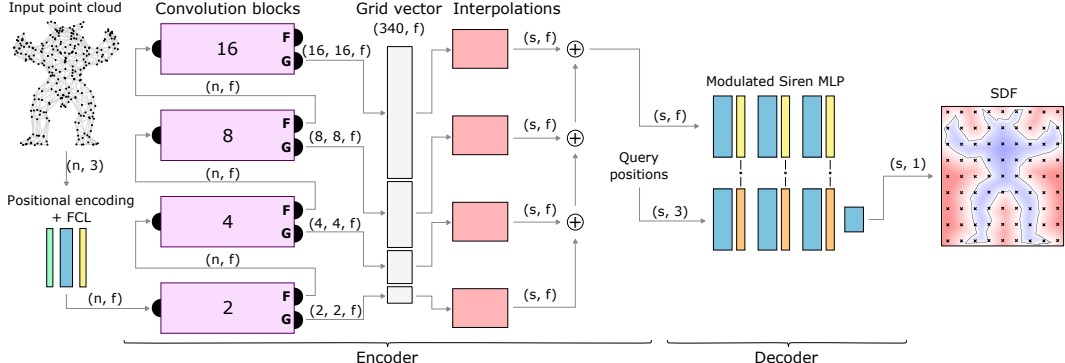

Figure 3: The encoder-decoder architecture of our network. The encoder computes vertex and volumetric features at multiple resolutions. By passing the feature vector through the convolution blocks, neighbor information is collected. The implementation of the convolution blocks is shown in Figure 2. After the last block, the vertex feature vector is discarded. The + denotes element-wise vector addition. $n$: number of vertices. $f$: number of features. $s$: number of SDF sample points.

where $\Omega$ is the SDFs domain and $\Omega_S$ is the surface of the object. Related to the work of Smith et al. (2021), Sitzmann et al. (2020b) proposed the following loss function as a measure of how well the Eikonal equation is satisfied

$$
\begin{aligned}
\mathcal{L}_{\text{eikonal}} = &\int_{\Omega} \|\,|\nabla\Phi(\boldsymbol{x})| - 1\| \; \mathrm{d}\boldsymbol{x} + \int_{\Omega_S} \|\Phi(\boldsymbol{x})\| \; \mathrm{d}\boldsymbol{x} \\
&+ \int_{\Omega_S} \left(1 - \langle\nabla\Phi(\boldsymbol{x}), \boldsymbol{n}_x\rangle\right) \mathrm{d}\boldsymbol{x} + \int_{\Omega\setminus\Omega_S} \exp(-\alpha|\Phi(\boldsymbol{x})|) \; \mathrm{d}\boldsymbol{x}.
\end{aligned}
\tag{2}
$$

Here $\Phi(\boldsymbol{x})$ denotes the predicted signed distance value of our neural network at position $\boldsymbol{x} \in \Omega$, and $\boldsymbol{n}_x$ denotes the target surface normal. The exponential in the last term is a weighting function which "pushes" signed distance values away from zero for points not on the surface. The original paper notes that the value of the constant $\alpha$ should be chosen as $\alpha \gg 1$, however we have found that this is sometimes detrimental to the desired effect, while choosing a lower value of $\alpha \approx 10$ yielded the best results.

If the input mesh is non-manifold or self-intersecting, the inside and outside of the object's volume are not well defined and normal vectors do not necessarily point in the right direction. For these cases we introduce a simple modification that ignores the sign of the normal:

$$
\mathcal{L}_{\text{surface normal}} = \int_{\Omega_S} \left(1 - |\langle\nabla\Phi(\boldsymbol{x}), \boldsymbol{n}_x\rangle|\right) \mathrm{d}\boldsymbol{x}.
\tag{3}
$$

As discussed in Section 4.3, this greatly improves performance on meshes with poor quality.

### 3.3 TRAINING

A major advantage of the proposed architecture is that ground truth SDF values for the input point-cloud never have to be computed. In order to evaluate the Eikonal loss only the sample position and classification into surface and non-surface samples is required.

We use three types of samples: 1) Surface points. If the source is a triangle mesh rather than a point-cloud, it is randomly sampled to create additional on-surface samples. 2) Random points in the volume are sampled as off-surface samples (no check is performed if these points actually lie on the surface per chance since the surface has zero measure as a volume). 3) If surface normal information is available, additional close-to-surface samples are generated from random points on the surface by displacing them along the surface normal. These additional samples are not strictly needed but aid the training process, as the SDF is most detailed close to the surface.

We train our encoder-decoder architecture end to end by inputting batches of surface point-clouds and sample points and computing output signed distance values. Automatic differentiation is used to provide spatial derivatives for the SDF gradients in $\mathcal{L}_{\text{eikonal}}$.

Table 1: A summary of the comparisons between our encoder method, ONet (Mescheder et al., 2019), ConvONet (Peng et al., 2020) and IF-Net (Chibane et al., 2020). The table shows the Chamfer distance (CD) and normal consistency (NC). The arrow indicates whether lower or higher values are better. The best score and all scores within 2% deviation are marked using a bold font.

| | Ours | | ConvONet | | IFNet | | ONet | |
|---|---|---|---|---|---|---|---|---|
| | CD ↓ mean / std $(\cdot 10^{-2})$ | NC ↑ mean / std $(\cdot 10^{-2})$ | CD ↓ mean / std $(\cdot 10^{-2})$ | NC ↑ mean / std $(\cdot 10^{-2})$ | CD ↓ mean / std $(\cdot 10^{-2})$ | NC ↑ mean / std $(\cdot 10^{-2})$ | CD ↓ mean / std $(\cdot 10^{-2})$ | NC ↑ mean / std $(\cdot 10^{-2})$ |
| Dragon | **2.455** / **0.168** | **96.25** / **0.754** | 3.960 / 1.788 | 88.89 / 7.987 | 2.681 / 0.305 | **95.63** / 1.480 | 5.581 / 2.408 | 82.20 / 8.904 |
| Armadillo | **0.956** / **0.063** | **97.86** / **0.591** | 2.377 / 1.840 | 90.80 / 5.827 | 1.259 / 1.678 | 95.87 / 2.529 | 4.451 / 2.667 | 83.35 / 6.613 |
| ShapeNet v2 | **2.011** / **0.409** | **86.11** / **3.731** | 3.734 / 1.933 | 75.43 / 6.454 | 2.180 / 0.513 | 82.34 / 3.942 | 34.89 / 3.142 | 59.61 / 4.807 |
| Thingi10k | **3.420** / **1.069** | **92.86** / **4.830** | 10.62 / 4.349 | 67.11 / 13.09 | 3.888 / **1.057** | **92.27** / 5.171 | 12.69 / 5.523 | 62.16 / 11.01 |
| Parameters | 747K | | 1.1M | | 1.9M | | 626K | |
| Inference | 2.34 s | | 0.7 s | | 15.3 s | | 2.45 s | |

## 4 RESULTS

The evaluation of our method is threefold: First, we perform an ablation study to motivate different design decisions in our architecture. Secondly, we compare different model sizes (i.e., number of learned parameters) with respect to the number of details they can reconstruct. And thirdly, we compare our results to previous encoder architectures to highlight the advantages of our approach.

**Datasets.** To cover a variety of relevant scenarios, we use four different datasets: The first two datasets consist of deformations of two different animated and deforming solid objects, a low-resolution dragon and a high-resolution armadillo, which were computed using the IPC simulation framework (Li et al., 2020). One of the core motivations of our work was being able to compute signed distances of deformable objects for collision detection. We have found this to be unsatisfactory with prior approaches, which is why we required our approach to be able to directly take any unprocessed surface mesh as input. To analyze performance on variable vertex count and connectivity, we employ the Thingi10k dataset as provided by Hu et al. (2018). Lastly, we use the "planes" category of the ShapeNetV2 dataset (Chang et al., 2015). As many meshes are of poor quality, we retriangulate them using Tetwild (Hu et al., 2018). However, this was not entirely sufficient to repair non-manifold meshes or fill holes in non-watertight geometry, so we use our proposed normal loss modification (see equation 3) for this dataset. For the last two datasets, we filter out some outliers with respect to vertex counts to keep the memory consumption reasonable. The number of vertices across all datasets ranged from 4 to 25,441. For more detail about our datasets, the reader is referred to Appendix A.3.

**Competing methods.** We focus our comparison on other encoder-based 3D shape representation methods, as they are most similar to our work. To that end we implemented the encoder components of Occupancy Networks (ONet) (Mescheder et al., 2019), Convolutional Occupancy Networks (ConvONet) (Peng et al., 2020) and Implicit Feature Networks (IFNet) (Chibane et al., 2020). To ensure a fair comparison, we use the same training and testing data for all models as well as the same modulated-SIREN decoder as discussed in Section 3. We use publicly available code to faithfully reproduce the models, while extending each network with the ability to handle arbitrary numbers of input points. When provided, we used the best configurations as suggested by the official code repositories, which resulted in the number of weights summarized in Table 1. For all comparisons with the other methods, our model utilizes five layers of latent grids with resolutions [4,8,16,32,64] and a latent size of 64. Furthermore, all options discussed in the ablation study in Section 4.1 are enabled.

Table 2: The results of comparing different design choices within the structure of our network. *Interpolation* refers to the grid interpolation, while *Graph Conv* and *Grid Conv* denote activation/ deactivation of all graph level and grid level convolutions respectively (see Figure 2). The table shows the Chamfer distance (CD) and normal consistency (NC). The data shown is for the dragon dataset. The cell color coding is derived from the mean value, different metrics use different color maps. A darker color corresponds to a better value.

| | Interpolation on | | | | Interpolation off | | | |
| --- | --- | --- | --- | --- | --- | --- | --- | --- |
| | Grid Conv on | | Grid Conv off | | Grid Conv on | | Grid Conv off | |
| | CD ↓ mean / std ($\cdot 10^{-2}$) | NC ↑ mean / std ($\cdot 10^{-2}$) | CD ↓ mean / std ($\cdot 10^{-2}$) | NC ↑ mean / std ($\cdot 10^{-2}$) | CD ↓ mean / std ($\cdot 10^{-2}$) | NC ↑ mean / std ($\cdot 10^{-2}$) | CD ↓ mean / std ($\cdot 10^{-2}$) | NC ↑ mean / std ($\cdot 10^{-2}$) |
| **Graph Conv** | | | | | | | | |
| on | 2.626 / 0.257 | 95.64 / 1.390 | 3.402 / 0.606 | 88.15 / 6.970 | 2.834 / 0.367 | 94.54 / 2.280 | 3.715 / 0.939 | 88.00 / 7.345 |
| off | 2.959 / 0.442 | 93.39 / 3.220 | 4.035 / 1.036 | 84.17 / 9.005 | 2.957 / 0.452 | 93.41 / 3.228 | 4.038 / 1.091 | 84.00 / 9.122 |

**Metrics.** We focus our evaluation on two different metrics: The widely used Chamfer distance (CD) and normal consistency (NC). The Chamfer distance compares distances between two point clouds $A$ and $B$ by finding the closest point in $B$ for each point in $A$ and taking the average. The normal consistency instead compares orientations of normals by also finding the closest point in $B$ for each point in $A$, but instead computing the cosine similarity of the normal vectors and taking the average. The same is done for the reverse direction and the results are summed together in the case of the Chamfer distance, and averaged in the case of the normal consistency. For all tests we use 200-400 individual reconstructions and report mean values and standard deviations for both metrics.

**Hardware.** We train all models on RTX 2080 Ti GPUs with 12GB of memory. With this, the training time for 400 epochs of our model using a batch size of 16 on a single GPU is around 1 day. All other models trained in similar time, apart from IFNet, which required roughly 4 days.

## 4.1 ABLATION

To highlight the impact of different design choices within our network architecture, we conduct an ablation study covering three decisions: 1) using the nearest neighbor instead of linear interpolation to map values from latent grid back to the mesh nodes (Figure 2), 2) enabling/ disabling the graph convolutions (Figure 2), and 3) enabling/ disabling the grid convolutions (Figure 2).

The results for different combinations are reported in Table 2. The most impactful component is clearly using the grid convolutions, which is also not surprising, because it contains the majority of trainable weights. The next important decision is enabling the graph convolution. Interestingly, the impact of enabling graph convolutions is greater when interpolation is also enabled. It also seems, that interpolation only has an impact on network performance when the graph convolution is also enabled. A visual comparison of Table 2 is shown in the Appendix in Figure 6. For maximal performance, all options should be turned on.

## 4.2 MODEL SIZE

The number of features per vertex and the resolutions of the grids are tunable parameters of our architecture. Increasing them enables the network to capture more surface details, at the cost of increased storage requirements or computation time. The results are shown in Figure 4. We find that the network behaves very intuitively and results degraded gracefully when the model size is reduced. Most notably, the models in the first row with a latent size of 16 all contain only 38K network parameters in total, of which the decoder contains just 1.4K. High-frequency features vanish first which makes small model sizes particularly appealing for applications such as approximate collision tests, e.g., in physical simulation. For a metric comparison of this experiment the reader is referred to Table 4 in the Appendix.

Increasing grid size

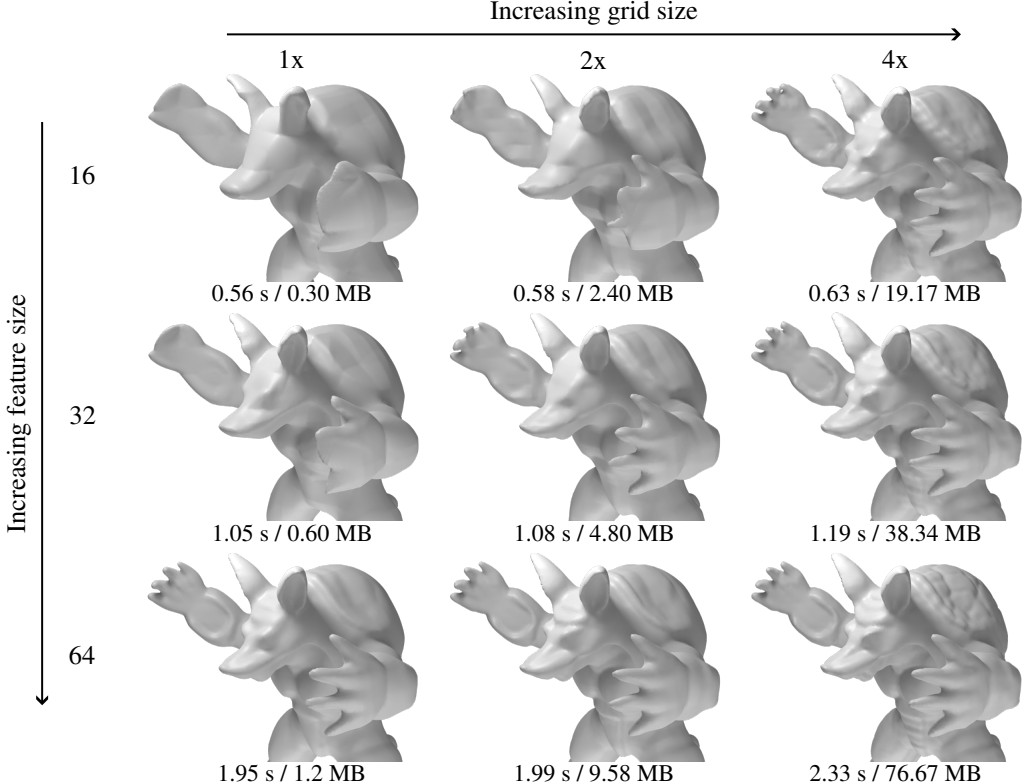

Figure 4: A comparison of reconstructed surfaces using our method. From left to right we increase the size of all latent grids, while from top to bottom the size of the latent feature is increased. Below each figure the inference time for 17M points (sampled on a $256^3$ regular grid) and the storage requirements for the latent grid is shown. The number of trainable network parameters for feature sizes 16, 32, and 64 are 38K, 151K, and 604K, respectively.

## 4.3 COMPARISON TO RELATED WORK

We now discuss comparisons to other state-of-the-art shape encoder methods on all four datasets. They are shown for all methods and datasets in the accompanying video, as well as in Table 1 and Figure 5. We can summarize that our method outperforms the other baselines with respect to reconstructed surface detail, both in terms of the Chamfer distance and normal consistency. Only for the Dragon and Thingi10k dataset, was the normal consistency of IFNet within 2% of our method. However, for these datasets the visual comparison in Figure 5 shows that our method is able to capture more detailed surfaces overall. Note especially the spine and claws of the dragon, and the distinct teeth on the cog-wheel. The cog-wheel unfortunately also shows a potential drawback of using k-nearest neighbor connectivity, where fine detail might become "connected". This is however difficult to remedy when only working with point-clouds. Nevertheless, our architecture is still able to capture fine details better than all other baselines.

Our method also is able to reconstruct a dense grid of approx. 17M points in only 2.35 s, while IFNet takes more than 6x longer. Only ConvONet is faster on the reconstruction task, however at significantly lower quality. We also find that our approach offers the highest quality per parameter.

Our model also has consistently very low standard deviation on all metrics, which makes the performance of the network more predictable. This is underlined by the comparison in Figure 4, which shows the effect of changing latent size and grid resolutions. Already very small networks with our architecture can have remarkable representational abilities.

Finally, we introduced a simple modification to the loss function to be able to compute signed distance fields for surface meshes with inconsistent surface normals, which we tested on the ShapeNet

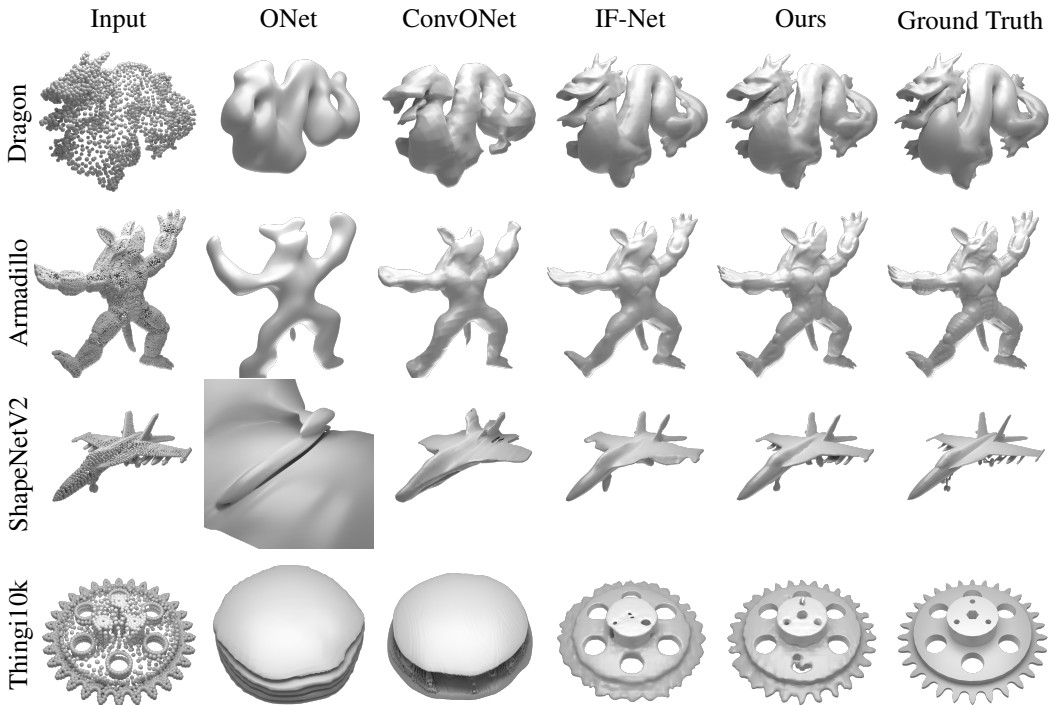

Figure 5: Comparing our method with related methods on different datasets. The reader is referred to the accompanying video for a more immersive comparison.

dataset. We observe an improvement on all metrics for three of the four methods, with the exception of ONets (Mescheder et al., 2019), which partially performed better using the original loss. Although it should be noted that the model did not seem to be able to deal well with the larger number of changing vertices in the dataset, which is why we interpret this result as an outlier. The overall metric improvement is shown in the Appendix in Table 5.

## 5    CONCLUSION AND FUTURE WORK

We have shown that our hybrid approach of interleaving mesh and grid convolutions, including back and forth projections, is capable of outperforming other state-of-the-art encoders on the surface reconstruction task from point cloud inputs. We are able to do this using only surface information and additional unlabeled samples in the surrounding volume, which prevents us from having to solve for a ground truth signed distance field in advance.

We believe that this work could be foundational for further research regarding efficient models for 3D shape encoding. For instance, exploring the sparsity of our latent grids could enable the usage of ever higher latent resolutions, resulting in accurate encoding of almost arbitrarily complex shapes. Combined with small latent sizes, this could result in an more scalable architecture, where the accuracy can be determined by available compute capabilities. Another interesting direction of further study is the use of the latent grid as a basis for editable neural fields. Multiple latent representations could be blended, or operators could be learned on the latent codes to achieve specific effects. Finally, in the spirit of using traditional geometric techniques within neural architectures, it could be explored whether projecting surface features to outer grid cells using methods such as fast marching or Laplacian smoothing could further improve predictions at distant points.

Code and data will be made available on our project website after publication.

## REPRODUCIBILITY STATEMENT

The general details of our model architecture are explained in Section 3.1. The loss function is presented in Section 3.2 and the training procedure in Section 3.3. The specific selection of network hyperparameters is discussed in Section 4.1, while other significant network parameter choices are given in paragraph "Competing methods" of Section 4. For a reference implementation regarding network architecture and data processing the reader is referred to the code on our project website.

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

## A APPENDIX

### A.1 FURTHER TRAINING DETAILS

We have implemented all of the models in this paper in the PyTorch framework (Paszke et al., 2019). In addition, we make use of the PyTorch-Geometric framework for its 3D graph and set learning capabilities (Fey & Lenssen, 2019). All models were trained using the Adam optimizer using a learning rate of 5e-4 and a batch size of 16. We would also like to note that we have implemented the tri-linear interpolation using custom CUDA kernels. This includes the double backward pass, which is not supported by the native PyTorch interpolation function, but necessary because we are using network gradients in our training target.

Table 3: Metadata about each of our datasets.

|  | Dragon | Armadillo | ShapeNet | Thingi10k |
|---|---|---|---|---|
| Train Meshes | 2400 | 2400 | 1632 | 2000 |
| Test Meshes | 300 | 300 | 421 | 200 |
| Vertices (min/max) | 2210 | 25441 | (434, 14879) | (4, 4995) |
| Faces (min/max) | 4220 | 50878 | (864, 30973) | (3, 10330) |

Table 4: Results of the resolution study. This table show the Chamfer distance (CD) and normal consistency (NC) for Figure 4. The cell color coding is derived from the mean value, different metrics use different color maps. A darker color corresponds to a better value.

| Latent | 16 | | 32 | | 64 | |
|---|---|---|---|---|---|---|
| | CD $\downarrow$ | NC $\uparrow$ | CD $\downarrow$ | NC $\uparrow$ | CD $\downarrow$ | NC $\uparrow$ |
| Grid | mean / std ($\cdot 10^{-2}$) | mean / std ($\cdot 10^{-2}$) | mean / std ($\cdot 10^{-2}$) | mean / std ($\cdot 10^{-2}$) | mean / std ($\cdot 10^{-2}$) | mean / std ($\cdot 10^{-2}$) |
| [2,4,8,16] | 2.163 / 0.524 | 88.87 / 3.071 | 1.828 / 0.466 | 90.72 / 3.001 | 1.383 / 0.357 | 94.35 / 2.707 |
| [4,8,16,32] | 1.415 / 0.135 | 92.45 / 1.416 | 1.266 / 0.127 | 93.95 / 1.616 | 1.098 / 0.110 | 96.24 / 1.421 |
| [8,16,32,64] | 1.052 / 0.049 | 96.18 / 0.804 | 0.987 / 0.054 | 97.20 / 0.687 | 0.946 / 0.047 | 97.90 / 0.576 |

## A.2 ADDITIONAL EVALUATION

Extending the evaluation of the main paper, we measure the reconstruction quality with an additional metric, which we call "surface point" loss. It is derived from the second term of Equation 2:

$$\mathcal{L}_{\text{surface point}} = \frac{1}{n} \sum_{i=0}^{n} ||\Phi(\boldsymbol{x}_i)||, \qquad (4)$$

for $\boldsymbol{x}_i \in \Omega_S$ being the input points on the zero level set. In essence, this metric computes the average distance from the target zero level set, to the predicted zero level set. In other words, it computes on average how far away the predicted surface is from the target surface, at each point on the target surface. Results are shown in Table 5. We find that our method consistently outperforms other methods when comparing surface point losses.

Table 5 also shows a column "Ours-mesh" in addition to "Ours-knn". The metrics in this column were obtained by using the surface mesh connectivity, instead of k-nearest neighbor connectivity of the vertices which was used for the main content of the paper. It can be seen, that both metics are generally within 2% of each other (denoted by the bold font), while using "ground-truth" surface mesh connectivity often (but not always) slightly improves the results.

## A.3 DATASETS

Table 3 shows additional meta data about the datasets. For all datasets, we pre-sample 100K sample points for each input mesh using the methodology described in Section 3.3. During each epoch, 30K samples are randomly drawn from the pre-sampled set for each mesh and used to evaluate the Eikonal loss.

|  | Interpolation on | | Interpolation off | |
|---|---|---|---|---|
|  | Grid Conv on | Grid Conv off | Grid Conv on | Grid Conv off |
| Graph Conv on | | | | |
| Graph Conv off | | | | |

Figure 6: A comparison of rendering for the ablation study conducted in the main document in Table 2. The ordering is the same as the presented table. The renderings clearly show, that the best results are achieved using both graph and grid convolution components *with* interpolation active. As soon as any components are disabled, there is a very clear loss of detail, even though the metrics do not change by a huge margin.

Table 5: Our method compared against ONet (Mescheder et al., 2019), ConvONet (Peng et al., 2020), and IFNet (Chibane et al., 2020). This is an extended version of Table 1 from the main paper. The table shows the Chamfer distance (CD), relative error (rel.) and the surface point loss (SP, see Section A.2). Due to the poor mesh quality of ShapeNet, a second version with adjusted loss function was trained (see Equation 3). Also, the relative error for the ShapeNet evaluation does not take the sign into account. For all three metrics, lower scores are better.

| | Ours-mesh | | | Ours-knn | | | ConvONet | | | IFNet | | | ONet | | |
| | CD $\downarrow$ mean/std ($\cdot 10^{-2}$) | NC $\uparrow$ mean/std ($\cdot 10^{-2}$) | SP $\rightarrow$ mean/std ($\cdot 10^{-2}$) | CD $\downarrow$ mean/std ($\cdot 10^{-2}$) | NC $\uparrow$ mean/std ($\cdot 10^{-2}$) | SP $\rightarrow$ mean/std ($\cdot 10^{-2}$) | CD $\downarrow$ mean/std ($\cdot 10^{-2}$) | NC $\uparrow$ mean/std ($\cdot 10^{-2}$) | SP $\rightarrow$ mean/std ($\cdot 10^{-2}$) | CD $\downarrow$ mean/std ($\cdot 10^{-2}$) | NC $\uparrow$ mean/std ($\cdot 10^{-2}$) | SP $\rightarrow$ mean/std ($\cdot 10^{-2}$) | CD $\downarrow$ mean/std ($\cdot 10^{-2}$) | NC $\uparrow$ mean/std ($\cdot 10^{-2}$) | SP $\rightarrow$ mean/std ($\cdot 10^{-2}$) |
|---|---|---|---|---|---|---|---|---|---|---|---|---|---|---|---|
| Dragon | **2.483**/ **0.164** | **96.44**/ **0.498** | 0.318/ **0.054** | **2.455**/ 0.168 | **96.25**/ 0.754 | **0.288**/ 0.057 | 3.960/ 1.788 | 88.89/ 7.987 | 1.226/ 0.956 | 2.681/ 0.305 | **95.63**/ 1.480 | 0.469/ 0.160 | 5.581/ 2.408 | 82.20/ 8.904 | 1.574/ 0.639 |
| Armadillo | **0.947**/ **0.047** | **97.86**/ **0.601** | **0.156**/ **0.027** | 0.956/ 0.063 | **97.86**/ **0.591** | 0.161/ 0.027 | 2.377/ 1.840 | 90.80/ 5.827 | 1.010/ 0.929 | 1.259/ 1.678 | 95.87/ 2.529 | 0.359/ 0.808 | 4.451/ 2.667 | 83.35/ 6.613 | 1.415/ 0.635 |
| ShapeNet v2 | 4.223/ 1.748 | 75.30/ 4.602 | 0.348/ 0.056 | 5.237/ 1.982 | 73.06/ 5.870 | **0.272**/ **0.048** | 4.424/ 1.969 | 72.31/ 5.290 | 0.896/ 0.485 | **2.724**/ **0.523** | **81.61**/ **3.709** | 0.505/ 0.097 | 10.02/ 4.116 | 58.11/ 4.148 | 0.510/ 0.261 |
| ShapeNet v2 (fixed) | **1.995**/ **0.406** | **86.44**/ **3.582** | 0.264/ 0.067 | 2.011/ 0.409 | 86.11/ 3.731 | **0.268**/ **0.059** | 3.734/ 1.933 | 75.43/ 6.454 | 0.965/ 0.720 | 2.180/ 0.513 | 82.34/ 3.942 | 0.399/ 0.125 | 34.89/ 3.142 | 59.61/ 4.807 | 1.435/ 0.634 |
| Thingi10k | 3.436/ 1.116 | **94.19**/ **3.738** | **0.266**/ **0.101** | **3.420**/ 1.069 | 92.86/ 4.830 | 0.276/ 0.103 | 10.62/ 4.349 | 67.11/ 13.09 | 2.138/ 1.311 | 3.888/ **1.057** | 92.27/ 5.171 | 0.754/ 0.225 | 12.69/ 5.523 | 62.16/ 11.01 | 1.184/ 0.625 |

