# OpenReview forum: "Zero-Level-Set Encoder for Neural Distance Fields"
_ICLR.cc/2024/Conference — ICLR 2024 Conference Withdrawn Submission_

### Official Review · Reviewer_ysCV · 2023-10-17

**Soundness:** 3 good
**Presentation:** 3 good
**Contribution:** 1 poor
**Rating:** 3
**Confidence:** 5

**Summary:**

This paper introduces an encoder/decoder neural network designed to predict Signed Distance Functions (SDFs). This network is trained using the Eikonal equation, eliminating the need for ground truth SDF supervision. The pipeline takes a 3D meshe as input and subsequently outputs the SDF value at any specified query point.

**Strengths:**

No training ground truth SDFs are required, which saves a bit of preprocessing computations. The network is trained using the Eikonal equation, eliminating the need for densely-sampled and accurate signed distances during training (which can nonetheless be obtained very easily, see weaknesses).

The new encoder architecture uses a unique multi-scale hybrid system that combines graph-based and voxel-based components, integrating both mesh and grid convolutions with projections from the mesh to the grid, at multiple scales.

The paper provides a solution for cases where surface normals are not well-defined, which is a common challenge in 3D geometry: simply using an unoriented cosine similarity.

Writing is very clear.

**Weaknesses:**

My central and huge concern is about the utility of such a pipeline: it inputs a mesh, outputs its SDF. This function (computing an SDF) can quickly be performed without any learning based technique, using standard geometric computing librairies like IGL or trimesh.
All the information is already present in the mesh! Why use a network to learn it?
Overall, using a neural network for this introduces computational overhead (the network needs to be trained), complexity, un-explainability, approximations, and has no clear motivation.

From this stems another weakness: the comparison with other baselines is unfair, since Convoccnet, IFNet and ONet take pointclouds as inputs, not meshes. In other words, they reconstruct a surface from an incomplete input, while the proposed pipeline has access to a full mesh.

If the method was about robustly getting an SDF out of a poorly triangulated mesh, then the whole paper needs to be rewritten with this target in mind. This means that the introduction should clearly set this goal, and the experiment sections needs to be reworked in order to include experiments on broken meshes with different defects, on which standard libraries fail.

Alternatively, if the method is about a novel mesh encoder network, then the task and decoder need to be changed to something else than regressing an SDF - part segmentation, classification….

Finally, if the point is about demonstrating that an SDF can be learned without explicit supervision, only by solving the Eikonal equation: this has already been demonstrated in SAL and SAL++ (Atzmon et al., these references are missing). For the cases shown in this submission, this is a made up problem, since ground truth SDF values can easily be computed, and are even used in the network evaluation. In other words, this does not enable new applications.

**Questions:**

Mostly: Why use a neural network to replace a traditional pipeline?

How does the proposed method perform in scenarios with noisy or incomplete data (pointclouds instead of meshes)?
How does the computational efficiency of the proposed method compare to traditional methods, especially in large-scale applications?

---

> ### Author Response · Authors · 2023-11-17
> **Reply to Official Review by Reviewer ysCV**
>
> Thank you for the feedback on our paper. To address most of your concerns we would like to refer you to the [Authors Statement](). For completeness we have included a response to the second part of the second question here.
>
> ***
>
> > [Q2] How does the proposed method perform in scenarios with noisy or incomplete data (pointclouds instead of meshes)? How does the computational efficiency of the proposed method compare to traditional methods, especially in large-scale applications?
>
> Regarding the comparison to traditional methods - we assume this refers to traditional surface reconstruction methods - we may revisit in the future. While a direct comparison would be interesting, we wanted to focus on comparisons with neural approaches in this work.

---

### Official Review · Reviewer_Msao · 2023-10-30

**Soundness:** 2 fair
**Presentation:** 3 good
**Contribution:** 2 fair
**Rating:** 5
**Confidence:** 4

**Summary:**

This paper proposes to represent 3D geometry using neural networks, using an encoder-decoder architecture. Using this design, it primarily addresses the task of 3D shape reconstruction from meshes.
The major technical contribution is the "hybrid" encoder architecture. Given a mesh, the method uses (1) a graph convolutional encoder to extract per-vertex features; (2) a multi-resolution grid structure to accumulate features from the vertices on grid nodes.
The authors also leverages the Eikonal loss to learn the neural signed distance field, obviating the need for pre-computing the SDF values in the training data. Although this is also claimed as a major contribution, it has been widely used in the neural shape modeling literature as of 2023.
Experiments have validated some of the design choices (such as the interpolation scheme when aggregating the grid features), and have compared to a few recent baselines. In general, the proposed method does show superior performance in terms of local geometric details. However, quantitative results do not consistently surpass certain baselines and some more recent work should have been considered as baselines.

**Strengths:**

- The hybrid encoder architecture is an intuitive design that makes sense, and is clearly demonstrated. Since the input are meshes, leveraging the graph convolution to extract features is a clever design which can (intuitively) bring extra information about the surface than only using the grid structures as in previous work (e.g. ConvONet and IFNet).

- The reconstructed surfaces have good quality especially in terms of local geometric details. On the airplane examples (as in the supplementary video), the model also shoes good performance reconstructing thin and (pontentially non-manifold) structures such as the fin and wings.

**Weaknesses:**

- Motivation: First of all, I'm wondering what's the practical application of the proposed method. The method assumes a mesh as input and aims to reconstruct a signed distance function from it, which also represents the geometry. If we already have the geometry well represented by a mesh, why is it necessary to reconstruct an SDF from it at a cost of losing certain surface details? On the other hand, given a mesh, one can directly compute the (signed) distance function by computing the distance from the query point to the surface. What's the benefit of introducing a neural network?

- Technical technical contribution:
   - The Eikonal loss is considered as a major technical novelty, but it has been proposed for learning 3D shapes in (Gropp et al. 2020)
 "Implicit Geometric Regularization for Learning Shapes (IGR)", and widely adopted for rendering implicit geometry (e.g. NeuS [Wang et al. NeurIPS 2021], VolSDF [Yariv et al. NeurIPS 2021]) and other downstream tasks such modeling deformable shapes (e.g. SCANimate [Saito et al. 2021]). This hurts a major technical contribution of this paper. At least the IGR paper by Gropp et al. should be cited and discussed.
   - While the hybrid encoder is interesting, the whole pipeline is more of a straightforward combination of standard modules (as of 2023) such as the graph convolution encoder, the multi-resolution features (as in IF-Net and NGLOD), and Siren decoder (as in the SIREN paper by Sitzmann et al). While admittedly this shouldn't be a major weakness per se, it is crucial to have more thorough ablation experiments to validate the intuitive combination. Most importantly, the graph conv + grid conv encoder is the key contribution. What would happen if the graph conv is shut down and one only uses the traditional point-encoder by densely sampling points from the input mesh surface? What if one doesn't use grid projection+interpolation at all, and simply uses the interpolated feature at the query point's nearest point on the mesh surface? To me, such experiments are critical in validating the technical contributions, but are missing.

- Experiments.
   - First of all, all baseline methods are from 2020 and do not represent the state-of-the-art performance. For example, POCO (Boulch et al., CVPR 2022) can be considered as a stronger baseline model for reconstructing shapes.
   - In terms of model performance, the proposed method has significantly higher "relative error" than IFNet on all datasets and there lacks a sound explanation supported by experiments. Again, given a mesh, computing the *accurate* SDF is straightforward, but from table 1, the proposed method cannot reproduce this property, which thus undermines its potential in the applications.
   - (minor) Table 2 only reports numbers on the Dragons and states 'results on other datasets are similar' -- I'd recommend showing all the results to make this statement more convincing.

**Questions:**

- For the baselines in Sec. 4.3, how many points are sampled from the mesh surface before sending into the encoder?
- In page 6, "Competing methods" paragraph, it is stated that the baseline methods are equipped with the same SIREN decoder as used in the proposed method. Does this yield a better performance than the original version of these models using their own decoder?
- In page 5, paragraph below Eq. 2 states that the last two terms in Eq. 2 are redundant but can improve training. Is there experimental results that support this argument?

**Details Of Ethics Concerns:**

I do not have ethical concerns about this submission.

---

> ### Author Response · Authors · 2023-11-17
> **Reply to Official Review by Reviewer Msao (1/2)**
>
> We appreciate the critical feedback and would like to address your concerns about our paper in the following.
>
> ## Weaknesses
>
> > [W1] Motivation: First of all, I'm wondering what's the practical application of the proposed method. [...] On the other hand, given a mesh, one can directly compute the (signed) distance function by computing the distance from the query point to the surface. What's the benefit of introducing a neural network?
>
> See the [Authors Statement]().
>
> ***
>
> > [W2.1] The Eikonal loss is considered as a major technical novelty, but it has been proposed for learning 3D shapes [...] This hurts a major technical contribution of this paper. At least the IGR paper by Gropp et al. should be cited and discussed.
>
> See the [Authors Statement]().
>
> ***
>
> > [W2.2] While the hybrid encoder is interesting, the whole pipeline is more of a straightforward combination of standard modules (as of 2023) [...]. While admittedly this shouldn't be a major weakness per se, it is crucial to have more thorough ablation experiments to validate the intuitive combination. [...] To me, such experiments are critical in validating the technical contributions, but are missing.
>
> See the [Authors Statement]().
>
> ***
>
> > [W3.1] First of all, all baseline methods are from 2020 and do not represent the state-of-the-art performance. For example, POCO (Boulch et al., CVPR 2022) can be considered as a stronger baseline model for reconstructing shapes.
>
> Thank you for suggesting a more recent work as a baseline alternative. While we would also like to compare with this method, the additional complexity of attention-based interpolation of features makes this method more expensive than other baselines. This should not rule the method out per-se, but it definitely makes it harder to justify comparisons with this method, when performance is desirable. We may however revisit this suggestion.
>
> ***
>
> > [W3.2] In terms of model performance, the proposed method has significantly higher "relative error" than IFNet on all datasets and there lacks a sound explanation supported by experiments. Again, given a mesh, computing the *accurate* SDF is straightforward, but from table 1, the proposed method cannot reproduce this property, which thus undermines its potential in the applications.
>
> We believe this is somewhat of a misinterpretation of the presented results. Our method has "slightly" higher relative error than IF-Net on most datasets, especially when compared to the other baselines. The intuitive explanation that we provide in the paper, is that our grid is sparse by design and has less degrees of freedom at distances far from the surface, resulting in larger relative errors, even though the accuracy at the surface is clearly better (see Chamfer distance and surface point loss).
>
> Nevertheless, as we now also train only on point clouds, we have to rerun the evaluation and will have to select a more suitable metric for comparing point clouds, as the *accurate* SDF computation becomes non-trivial. Other related works suggest to use e.g. the Hausdorff distance or the normal consistency loss in addition to the Chamfer distance. We will update the metrics in main document once we obtain them.
>
> ***
>
> > [W3.3] (minor) Table 2 only reports numbers on the Dragons and states 'results on other datasets are similar' -- I'd recommend showing all the results to make this statement more convincing.
>
> The deforming dragons experiment is absolutely representative for the model performance during ablation, but a more complete picture could be provided by giving metrics for the other datasets in the future. We will keep this suggestion in mind for the future, thank you.

---

> > ### Comment · Reviewer_Msao · 2023-11-23
> >
> > Thanks for the clarifications but I'm still not sure if I can get the response to W3.2. It's true that the proposed model only takes point cloud as input, but that sounds to me more like a design choice to drop the mesh connectivity information -- since the application is mesh-to-SDF, the input to the whole system is a mesh, and computation of the accurate SDF is straightforward. Am I missing something?
> >
> > Also regarding W3.3, while I understand that intuitively the dragon experiment is representative as it is sufficiently challenging, I still believe that it is not sufficiently scientific to simply state that results on other datasets are similar without actually showing the results. If the experiments are done, the results can simply be shown in the final version.

---

> > > ### Author Response · Authors · 2023-11-23
> > >
> > > Thank you for taking the time to consider our response and our updated paper.
> > >
> > > > Thanks for the clarifications but I'm still not sure if I can get the response to W3.2. It's true that the proposed model only takes point cloud as input, but that sounds to me more like a design choice to drop the mesh connectivity information -- since the application is mesh-to-SDF, the input to the whole system is a mesh, and computation of the accurate SDF is straightforward. Am I missing something?
> > >
> > > Indeed we were not completely clear in the response to your review initially. The application is essentially **no longer** mesh-to-SDF but instead **point-cloud-to-SDF**. All mesh connectivity is disregarded before training starts, such that our method and all other baselines only have point information to work with. We apologize that this change was not clear in our original comment and we understand that after having read the original paper, this was **supposed to be** one of the main takeaways. This is no longer the case, as we now completely work on point data.
> > >
> > > > Also regarding W3.3, while I understand that intuitively the dragon experiment is representative as it is sufficiently challenging, I still believe that it is not sufficiently scientific to simply state that results on other datasets are similar without actually showing the results. If the experiments are done, the results can simply be shown in the final version.
> > >
> > > We absolutely agree, and are definitely willing to add the results of the ablation study for the other datasets to the appendix for a final version, but unfortunately are a bit pressed for resources and time to do so before the rebuttal deadline.

---

> ### Author Response · Authors · 2023-11-17
> **Reply to Official Review by Reviewer Msao (2/2)**
>
> ## Questions
>
> > [Q1] For the baselines in Sec. 4.3, how many points are sampled from the mesh surface before sending into the encoder?
>
> We apologize for the oversight of not mentioning this explicitly. We send all of the mesh vertices into the encoder. Along with using raw data for training and testing, we found it would be helpful to be able to have an architecture that is agnostic to the number of input points. We have extended the other baselines to be able to do the same, as their underlying methods, e.g. PointNet or occupancy-driven CNNs  are also agnostic to the number of input points. The same holds true for our presented network architecture.
>
> As we show in our results, and especially on the datasets with wildly varying vertex counts (ShapeNet and Thingi10k), it is quite possible to achieve good results using a variable number of input points. That being said, additional points for training are sampled in the manner described in Section 3.3 on page 5.
>
> ***
>
> > [Q2] In page 6, "Competing methods" paragraph, it is stated that the baseline methods are equipped with the same SIREN decoder as used in the proposed method. Does this yield a better performance than the original version of these models using their own decoder?
>
> It was in our opinion the fairest way to do the comparisons. As was shown in previous papers on using network derivatives, ReLU activations (usually employed in the other baseline decoders) typically perform worse than sin activations. While it might be possible to yield better results when training the baseline methods on very densely sampled ground truth signed distance values, as used to be done in the past, our method obviates the need for these values to be precomputed. We believe it was therefore not possible to use the decoders of the respective methods in a "fair" way, without compromising the comparison in one way or another.
>
> ***
>
> > [Q3] In page 5, paragraph below Eq. 2 states that the last two terms in Eq. 2 are redundant but can improve training. Is there experimental results that support this argument?
>
> In tests leading up to the solution for flipped surface normals, we have attempted to entirely remove the loss specifying the direction of the surface normal. This however resulted in less detailed and more "washed-out" surfaces.
> In early tests we have also observed, that removing the exponential term which "pushes" non-zero SDF values away from the surface, resulted in slightly noisier SDF values far away from the surface. In early tests, it could also cause convergence issues during training at times, yet at that time the architecture had not been finalized and the error could have originated from other sources. We will consider running experiments to support this claim but will remove it from the paper for the time being.

---

> > ### Comment · Reviewer_Msao · 2023-11-23
> >
> > Thanks to the authors for the detailed clarifications and the experimental results unlisted in the original submission. My questions are mostly addressed by these arguments and would suggest adding these nice arguments to the final version.

---

### Official Review · Reviewer_ruBo · 2023-11-02

**Soundness:** 3 good
**Presentation:** 3 good
**Contribution:** 2 fair
**Rating:** 5
**Confidence:** 4

**Summary:**

This paper proposes an efficient encoder-decoder architecture to encode 3D shapes as implicit neural signed distance fields. The core idea is to combine graph and voxel-based encoders coupled with an implicit decoder that can be trained using the Eikonal equation enforced on the shape boundary using surface samples without the need for computing signed distance values on the ground truth data. A modified loss function is also presented for handling meshes that are not watertight or have unoriented normals.

**Strengths:**

- The ability to fit a neural network to a shape without having access to ground truth signed distances values is a big strength.
- The method is conceptually simple and easy to understand, while being effective and efficient. It has been demonstrated on various datasets where it outperformed other chosen related work.
- The hybrid graph and voxel based encoder is interesting and novel, and could be significant for future research involving mesh encoding in general.

**Weaknesses:**

- One the main contributions of the paper is the hybrid graph and voxel based encoder, but it is not evaluated comprehensively.  An ablation study on completely removing the graph and voxel based components of the encoder would be useful in understanding the importance of this contribution.
- Some very relevant papers are missing in comparisons and related work. These works can also encode a shape into a neural field without having access to ground truth SDF values at the sample points:
  - SAL: Sign Agnostic Learning of Shapes from Raw Data (CVPR 2020)
  - Implicit Geometric Regularization for Learning Shapes (ICML 2020)
  - SALD: Sign Agnostic Learning With Derivatives (ICLR 2021)

**Questions:**

- How important are the individual graph and voxel components of the proposed encoder network, and the encoder network itself as a whole? An ablation study would be helpful to understand this contribution better.
- How does the proposed method compare against the missing related work listed above in the Weaknesses section? While not all of these works are encoder-decoder models, a comparison is necessary since they too share the advantage of the proposed method of not requiring ground truth SDF values at samples.

---

> ### Author Response · Authors · 2023-11-17
> **Reply to Official Review by Reviewer ruBo**
>
> Thank you very much for the insightful review. We are glad that the main points of our paper seem to have come across well. We have found both questions/ concerns to be valuable for improving our paper and discuss these in the [Authors Statement]().

---

> > ### Comment · Reviewer_ruBo · 2023-11-23
> >
> > Thanks for revised ablation study on the hybrid architecture and the discussion on the missing related works. My concerns have been largely addressed and I think these will be great additions to the final paper.

---

### Author Response · Authors · 2023-11-17
**Authors Statement (3/3)**

### Reviewer [ysCV](https://openreview.net/forum?id=BC4AUywMow&noteId=p9wGGmVNJ2)

> Finally, if the point is about demonstrating that an SDF can be learned without explicit supervision, only by solving the Eikonal equation: this has already been demonstrated in SAL and SAL++ (Atzmon et al., these references are missing)

### Reviewer [Msao](https://openreview.net/forum?id=BC4AUywMow&noteId=bsDIwzV0FZ)

> The Eikonal loss is considered as a major technical novelty, but it has been proposed for learning 3D shapes [...] This hurts a major technical contribution of this paper. At least the IGR paper by Gropp et al. should be cited and discussed.

### Reviewer [ruBo]((https://openreview.net/forum?id=BC4AUywMow&noteId=NugdJvBwDN))

> How does the proposed method compare against the missing related work listed above in the Weaknesses section? While not all of these works are encoder-decoder models, a comparison is necessary since they too share the advantage of the proposed method of not requiring ground truth SDF values at samples.
>  - SAL: Sign Agnostic Learning of Shapes from Raw Data (CVPR 2020)
>  - Implicit Geometric Regularization for Learning Shapes (ICML 2020)
>  - SALD: Sign Agnostic Learning With Derivatives (ICLR 2021)

### Architectures

We thank the reviewers for making us aware of these works, we will update the paper in the following days, such that we discuss these papers appropriately, as they also solve a very similar problem. Indeed, the formulations of the loss in IGR and SALD have similarities with our formulation and we were very curios to read about the solution to flipping signs in the SAL paper. It would also be interesting for future work, whether the specialized weight initialization scheme could be combined with the one presented for SIREN networks.

Nevertheless, none of the presented architectures are suitable for direct comparison with our method, as all of them rely on a similar auto-decoder structure as was initially presented in DeepSDF [Park et al. 2019]. An exception is the variational encoder-decoder presented in SAL and SALD, which is however also (optionally) augmented by an optimization loop to improve the results. Since the encoder of SAL is claimed to be PointNet (Qi et al. 2017), the results will likely be very similar to Occupacy Networks (which we do compare with). In fact, the authors of SAL and SALD explicitly state that their encoder architecture is similar to the one used by Mescheder et al. 2019 for Occupancy Networks. That is as long as the optimization loop is not employed.

As one of our main goals was to have an **efficient** architecture for reconstructing and querying SDFs, we ruled out comparing with auto-decoder methods, as they rely on optimizing latent codes during inference. As such, they are typically able to reconstruct high fidelity SDFs at the upfront cost of computation time as well as computationally expensive SDF queries. To put things into perspective, the named papers all need to evaluate an 8-layer MLP decoder with 256 to 512 hidden units to compute a single signed distance. In contrast, we need to perform one tri-linear interpolation and evaluate 3 layers with 64 hidden units per SDF query, once the latent grid has been computed. While the computation of our latent grid is not free, it is typically orders of magnitude faster than querying all SDF values for surface reconstruction.

In practice, auto-decoders often also have reduced test sets, as evaluation on new inputs is relatively expensive compared to encoder-decoder approaches. This is also corroborated by e.g. the suggested related work of Reviewer [Msao](https://openreview.net/forum?id=BC4AUywMow&noteId=bsDIwzV0FZ), POCO [Boulch and Marlet 2022].

### Eikonal loss

Regarding the Eikonal loss for learning SDFs in an unsupervised manner we would like to clarify the following. While previous works demonstrated the ability to learn an SDF without supervision by using the Eikonal equation, it has yet to be demonstrated in a pure forward-only encoder-decoder setting to high fidelity. Other works, including IGR by Gropp et al. to the best of our knowledge, make use of latent code optimization at test time, while our network directly computes valid latent codes in a single forward pass. This means our method is inherently more efficient in adapting to new shapes, as we forego the costly optimization loop during inference. If nothing else, our method takes a step in the direction of making the Eikonal loss more usable as a training target, and neural distance fields as a whole more usable for practical and performance critical applications.

---

### Author Response · Authors · 2023-11-17
**Authors Statement (2/3)**

> How does the proposed method perform in scenarios with noisy or incomplete data (pointclouds instead of meshes)?

We acknowledge that for the way our paper was written and considering the goals of compared related works, that it may have seemed a bit nonsensical to compute the SDF from a mesh, which can be obtained reasonably efficiently from the mesh data. Therefore, we have taken the suggestion into account and are rerunning all results using only point-data for our method as well. In other words, we **completely discard the mesh connectivity** and only use it to sample additional points for training. As mentioned in the method section of the paper, we simply construct a k-nearest neighbor graph and use this for graph convolutions instead of the mesh connectivity. Similar to related works, we now reconstruct SDFs directly from point data. We are seeing promising initial results using this pipeline and will update the paper once we obtain a better picture.

By using point-clouds instead of surface meshes, we are able to present a more general framework. One can interpret using surface meshes, as was done originally, as a specialization of this framework. In this way, we show both a fairer comparison to other baselines, i.e. all methods now use only point data, while also expanding the usability of our method.

***

## Ablation

Another point that Reviewer [ruBo](https://openreview.net/forum?id=BC4AUywMow&noteId=NugdJvBwDN) and Reviewer [Msao](https://openreview.net/forum?id=BC4AUywMow&noteId=bsDIwzV0FZ) seemed to agree on, is that the contribution of the novel hybrid encoder architecture was impacted by not providing more thorough ablation, or rather more relevant cases in the ablation study.

### Reviewer [Msao](https://openreview.net/forum?id=BC4AUywMow&noteId=bsDIwzV0FZ)

> While the hybrid encoder is interesting, the whole pipeline is more of a straightforward combination of standard modules (as of 2023) [...]. While admittedly this shouldn't be a major weakness per se, it is crucial to have more thorough ablation experiments to validate the intuitive combination. [...] To me, such experiments are critical in validating the technical contributions, but are missing.

### Reviewer [ruBo](https://openreview.net/forum?id=BC4AUywMow&noteId=NugdJvBwDN)

> How important are the individual graph and voxel components of the proposed encoder network, and the encoder network itself as a whole? An ablation study would be helpful to understand this contribution better.

The motivation for doing the ablation study in the way it was originally presented, was to keep the network complexity, i.e. number of parameters, identical across all runs. This would highlight the importance of specific architectural design decisions, such as interpolating grid features back to the input vertices. Otherwise, it would be difficult to tell if an improvement came from having an increased number of network parameters available to train, or from the actual way of interconnecting nodes.

Nevertheless, as there seems to be genuine interest in the investigation of architectural variations, we are currently running an updated ablation study. We will update the paper with the results once the runs finish. We may opt to move the previous ablation into the appendix to preserve it for reference. To the reviewers further credit, this seems to indeed yield some very interesting insights into the effect when using only the voxel and graph components, with or without interpolation.

***

## Related Work

Finally, all Reviewers [ruBo](https://openreview.net/forum?id=BC4AUywMow&noteId=NugdJvBwDN), [Msao](https://openreview.net/forum?id=BC4AUywMow&noteId=bsDIwzV0FZ) and [ysCV](https://openreview.net/forum?id=BC4AUywMow&noteId=p9wGGmVNJ2) pointed out relevant, but missing, related works. Since these works had a lot of overlap between reviewers, we would like to address them jointly here.

---

### Author Response · Authors · 2023-11-17
**Authors Statement (1/3)**

First of all we would like to thank all the reviewers for taking the time to read and understand our manuscript, as well as to provide us with valuable feedback. We will use this feedback to make improvements to the paper in the following days. We will make another post once we have updated the paper.

In the following we want to address concerns raised by multiple reviewers, while the responses to individual reviewers will be posted as direct comments.

***

## Motivation

The biggest weakness raised by both Reviewer [ysCV](https://openreview.net/forum?id=BC4AUywMow&noteId=p9wGGmVNJ2) and Reviewer [Msao](https://openreview.net/forum?id=BC4AUywMow&noteId=bsDIwzV0FZ), is the lackluster motivation of using a mesh as input to a neural network and computing the signed distance as output.

### Reviewer [ysCV](https://openreview.net/forum?id=BC4AUywMow&noteId=p9wGGmVNJ2):

> My central and huge concern is about the utility of such a pipeline: it inputs a mesh, outputs its SDF. This function (computing an SDF) can quickly be performed without any learning based technique, using standard geometric computing librairies like IGL or trimesh. Overall, using a neural network for this [...] has no clear motivation.

>  Mostly: Why use a neural network to replace a traditional pipeline?

### Reviewer [Msao](https://openreview.net/forum?id=BC4AUywMow&noteId=bsDIwzV0FZ):

> Motivation: First of all, I'm wondering what's the practical application of the proposed method. [...] On the other hand, given a mesh, one can directly compute the (signed) distance function by computing the distance from the query point to the surface. What's the benefit of introducing a neural network?

We believe that this is the main point of miscommunication in our paper. Our original motivation for researching in this direction was to use neural networks for capturing deforming rigid objects (see deforming dragon and armadillo datasets). We wanted to create a utility to assist in fast (but approximate) collision detection and collision handling for solid-object simulations, something for which SDFs have been found to be useful in previous research on the topic:

- Xu and Barbič. “Signed Distance Fields for Polygon Soup Meshes.” 2014
- Koschier et al. “An Hp-Adaptive Discretization Algorithm for Signed Distance Field Generation.” 2017
- Macklin et al. “Local Optimization for Robust Signed Distance Field Collision.” 2020

SDFs are very attractive for determining collisions of solid objects, as an explicit map can be computed to a desired resolution ahead of time and queried **much** faster, than computing explicit triangle-point distances. If the object is deforming however, SDFs are not typically used, as this map needs to be updated very frequently, which can become quite costly for interactive applications, where a single time-step is only allowed to take ~40ms including physical solvers. This is where the idea for this work originated, i.e. would it be possible to use neural networks to accelerate updating approximate SDF maps and to enable fast SDF queries for deformable objects? While computing distances from a mesh is a solved problem, it needs to scale appropriately to be usable in the context of solid-object simulations. We believe that a neural network with an efficient encoder and an "as-small-as-possible" decoder could take good strides in that direction. This is also why we discarded auto-decoder approaches as untenable, since even a single optimization loop during inference already breaks computational constraints.

The previous paragraph should simply provide context to the original motivation of our work. In the process, we found that our architecture also worked incredibly well on other datasets and the general task of encoding 3D geometry into neural fields. In the process of shifting focus, we have simply not re-evaluated the idea of using meshes. As such, we truly appreciated the following suggestion of Reviewer [ysCV](https://openreview.net/forum?id=BC4AUywMow&noteId=p9wGGmVNJ2) to train our network on point-clouds instead, as this makes for an even more generally usable architecture.

---

### Author Response · Authors · 2023-11-21
**Rebuttal Revision**

We have now uploaded a rebuttal revision of our paper as well as an updated video as supplemental material. We have made the following changes, to address all major concerns about the paper:

- re-ran all experiments of our model using only point-cloud inputs, and updated all figures and metrics. This addresses major concerns about the motivation and usefulness of our work and the fairness of comparisons to other baselines. We also adjusted the text to be consistent with the new experiments and pipeline.
- cited the requested related work and discussed it in an additional brief paragraph.
- re-ran the ablation study to cover more interesting variations of our architecture. We have also rendered the results of different variations in the appendix for visual comparison. If interest remains in the original ablation study, we would be willing to add this to the appendix for the camera ready version.
- improved the selection of comparison metrics for evaluation on point clouds.

We hope that these changes find the approval of the reviewers and we stand by for any further questions or comments.

---

> ### Comment · Reviewer_Msao · 2023-11-23
>
> I appreciate the updated version with additional results, ablations and discussions of related work. The new version looks definitely more solid than the original submission. The Author Statement in the rebuttal has clarified the motivation behind the work.
>
> However I do still have a concern. Now that the method has a big change than the original submission version (i.e. the assumed input to the pipeline is now a pure point cloud without connectivity information; originally it is a mesh), I'm wondering the importance of the graph convolution module in the pipeline, as this is the key differentiator in the encoder architecture from the baselines, e.g. ConvONet or IF-Net. Since the connectivity for the graph convolution is constructed using KNN search (in the updated paper version), I'm wondering, how much additional information does the graph conv module introduce than the grid convolution module? In other words, does the performance gain brought by the graph conv module in the new Table 2 because of the extra network parameters it introduces, or by the connectivity information that it leverages?
>
> A concrete counter-example is when applying the method on articulated data, e.g. human body (as used in many relevant works, e.g. SAL and IGR). When a human has a cross-arm pose, the knn graph construction would associate a point from the hand to the torso. But intrinsically the geodesic distance on the body manifold between the hand and torso should remain large no matter the pose. In this conflicting example, would the graph conv module in the proposed method help, or hurt?
>
> Based on my mixed observations and thoughts above, I would raise my rating to a boarderline score.

---

> > ### Author Response · Authors · 2023-11-23
> >
> > These are excellent questions and something that we can only answer by deduction from multiple of the key results of our paper.
> >
> > ***
> >
> > > Since the connectivity for the graph convolution is constructed using KNN search (in the updated paper version), I'm wondering, how much additional information does the graph conv module introduce than the grid convolution module? In other words, does the performance gain brought by the graph conv module in the new Table 2 because of the extra network parameters it introduces, or by the connectivity information that it leverages?
> >
> > The fact that additional parameters typically result in a metric improvement is the reason for our original ablation study design. Indeed, purely from the metrics it is difficult to discern whether additional improvements come from having a larger number of parameters available or from the clever construction of our architecture.
> >
> > Nevertheless, we would argue that especially the comparison with IFNet shows that our usage of graph (or point) convolutions indeed achieves greater accuracy, especially with respect to fine details. IFNet uses a dense UNet structure (i.e. pure 3D CNN) with almost 3 times as many parameters as our network, yet is still not able to reconstruct the geometry with as high fidelity as our method is. We use fewer parameters including our graph network, and are able to obtain better results. Compounding this we can see in our ablation study that using only grid convolutions is not sufficient to capture fine details of the geometry (Figure 6 in the Appendix). Overall, there is a huge number of other variations one could try and we willing to provide more comparisons in case this argumentation is not convincing.
> >
> > Finally, it would **probably** also be possible to achieve similar results using purely a CNN architecture, given enough parameters, a fine enough grid size and enough computational resources. However, we have not managed to find this configuration in our testing. Therefore it is more of a tradeoff: Using our architecture is **more efficient and predictable** with regard to the captured detail, in terms of runtime and, also very crucially in 3D deep learning, lower GPU memory usage.
> >
> > ***
> >
> > > A concrete counter-example is when applying the method on articulated data, e.g. human body (as used in many relevant works, e.g. SAL and IGR). When a human has a cross-arm pose, the knn graph construction would associate a point from the hand to the torso. But intrinsically the geodesic distance on the body manifold between the hand and torso should remain large no matter the pose. In this conflicting example, would the graph conv module in the proposed method help, or hurt?
> >
> > Unfortunately, we cannot show results on this dataset. While we believe that our method would still perform very well, we cannot support such a claim. It could be possible that using normal orientations becomes crucial for these kinds of cases, or that the network simply learns to discard these kinds of connections by combining information about the local and global context. We are definitely looking at incorporating this dataset for future work.
> >
> > ***
> >
> > We appreciate the work you have put in to understand our work and provide insightful feedback, thus enabling us to make numerous improvements to the paper.